# Medicinal Use of Testosterone and Related Steroids Revisited

**DOI:** 10.3390/molecules26041032

**Published:** 2021-02-15

**Authors:** Jan Tauchen, Michal Jurášek, Lukáš Huml, Silvie Rimpelová

**Affiliations:** 1Department of Food Science, Faculty of Agrobiology, Food, and Natural Resources, Czech University of Life Sciences Prague, Kamýcká 129, 165 00 Prague-Suchdol, Czech Republic; 2Department of Chemistry of Natural Compounds, University of Chemistry and Technology Prague, Technická 5, Prague 6, 166 28 Prague, Czech Republic; michal.jurasek@vscht.cz (M.J.); lukas.huml@vscht.cz (L.H.); 3Department of Biochemistry and Microbiology, University of Chemistry and Technology Prague, Technická 5, Prague 6, 166 28 Prague, Czech Republic; silvie.rimpelova@vscht.cz

**Keywords:** androgen receptor, antiandrogens, aging, longevity, medicinal natural products, performance-enhancing drugs, selective androgen receptor modulators, testosterone

## Abstract

Testosterone derivatives and related compounds (such as anabolic-androgenic steroids—AAS) are frequently misused by athletes (both professional and amateur) wishing to promote muscle development and strength or to cover AAS misuse. Even though these agents are vastly regarded as abusive material, they have important pharmacological activities that cannot be easily replaced by other drugs and have therapeutic potential in a range of conditions (e.g., wasting syndromes, severe burns, muscle and bone injuries, anemia, hereditary angioedema). Testosterone and related steroids have been in some countries treated as controlled substances, which may affect the availability of these agents for patients who need them for therapeutic reasons in a given country. Although these agents are currently regarded as rather older generation drugs and their use may lead to serious side-effects, they still have medicinal value as androgenic, anabolic, and even anti-androgenic agents. This review summarizes and revisits the medicinal use of compounds based on the structure and biological activity of testosterone, with examples of specific compounds. Additionally, some of the newer androgenic-anabolic compounds are discussed such as selective androgen receptor modulators, the efficacy/adverse-effect profiles of which have not been sufficiently established and which may pose a greater risk than conventional androgenic-anabolic agents.

## 1. Introduction

Testosterone (T) derivatives and their (semi-)synthetic analogues (so-called androgenic anabolic steroids—AAS) have been controversial for already quite some time. These substances have become the subject of abuse by professional athletes, and currently also by a significant number of amateur athletes, to enhance performance (i.e., performance-enhancing drugs) and body aesthetics. However, T and many AAS have valuable and often irreplaceable pharmacological activities that are medicinally useful, though these compounds are currently regarded as rather older generation drugs.

T and related compounds primarily act as androgens, promoting the development and maintenance of male sex characteristics such as maturation of the sex organs, voice deepening, and growth of facial and body hair. They also have an anabolic activity that promotes the storage of protein and stimulates the growth of bone and muscles, and these functions are especially important from a medicinal standpoint [1]. Indeed, tremendous efforts have been put into developing agents with increased anabolic activity such as the recently discovered selective androgen receptors modulators (SARMs). However, there is still no single anabolic molecule from which the androgenic activity has been fully eliminated. T and other AAS still find their use in the treatment of a wide range of human diseases, including hypogonadism, male sexual impotence, and some types of breast cancer in women. They are also of value in various types of wasting syndromes, for example in patients suffering from acquired immune deficiency syndrome (AIDS) anorexia, or alcoholism, and for those with severe burns, muscle, tendon or bone injury, osteoporosis, certain types of anemias, and hereditary angioedema [2]. T has also recently been discussed in connection with longevity. As a person ages, their physiological levels of T decrease. The T decline has been associated with aging symptoms such as hypertension, obesity, diabetes, overall fatigue, depression, and cognitive decline [2,3]. The current trend in some countries is to use T with its pleiotropic effects to combat several age-related changes, rather than a combination of drugs each treating one symptom. Supplementation with T or related compounds, however, may cause serious adverse effects, including skin disorders, hepatotoxicity (especially true for the orally-active T derivatives), altered blood lipid profiles, hypertension, cardiovascular conditions, kidney disorders, behavioral changes, and reproduction disorders [4]. Regardless of their safety and side-effect profile, T and its analogues, at the correct formulation and dose for the appropriate condition, may still offer several beneficial pharmacological responses and may be considered very valuable pharmaceutical agents.

One of the biggest current problems associated with AAS is that there has been an increasing number of recent reports of AAS abuse by non-professional athletes, mostly young people seeking to improve performance, build muscle and stamina, and have a great looking body [5,6]. Apart from the aforementioned side effects, AAS use may lead to withdrawal symptoms after these drugs have been discontinued. These symptoms are very similar to those observed in subjects with age-related T decline, including increased fat storage, loss of muscle mass and bone strength, mood swings, irritability, extreme fatigue, restlessness, and depression. Thus, for many users, the only way to overcome these symptoms is to start taking AAS again, and ultimately, they become addicted to these drugs (however, it is a relatively special type of addiction that is different from other drugs). As there are some indications that the abuse of AAS by amateur athletes is increasing, this may pose a challenge to the health care system and addiction centres.

This review principally aims to summarize particular examples of T analogues and other androgens, including established anabolic steroids, newly introduced SARMs, T- and nandrolone-prohormones, formulations containing steroids intended to raise endogenous T levels (so-called T boosters), and drugs that act as antiandrogens based on T structure and revisits their role as therapeutic drugs. The information summarized in this review was obtained through an extensive review of the literature by searching for relevant books and articles with the Web of Knowledge, SciVerse Scopus, and PubMed databases.

## 2. Available Testosterone Analogues

Medicinal application, routes of administration, and available forms of compounds discussed in this review are summarized in Table 1.

### 2.1. Analogues of Testosterone with Agonistic Activity

Androgens generally consist of a C_19_ androstane skeleton without a side chain and they have Δ^4^-3-keto and 17β-hydroxyl functional groups. The primary natural androgen is T (Figure 1 and Figure 2).

T is synthesized de novo from cholesterol via several enzymatic transformations where dehydroepiandrosterone (DHEA) **1**, androstenediol **2**, and androstenedione **3** play a key role (Figure 1). T may be subject to further structural changes leading to the production of dihydrotestosterone **4** or estradiol (Figure 1). T can be produced pharmaceutically from androstenolone (**5**, Figure 2) by the reduction of 17-carbonyl and oxidation of the 3-hydroxyl with the use of necessary protecting groups. This structural conversion is attained by the use of yeasts, which first oxidize the 3-hydroxyl under aerobic conditions, then reduce the 17-keto group under anaerobic conditions (Figure 2A). Androstenolone may be obtained from plant-derived steroids, such as diosgenin from *Dioscorea* species (Dioscoreaceae), *Trigonella foenum-graecum* (Fabaceae), and solasodine or tomatidine from various *Solanum* and *Lycopersicon* species (Solanaceae) by marker degradation and side-chain removal. Additionally, **5** can be further synthetically modified by alkylation at the C-17 position and successive oxidation resulting in potent anabolics 17α-methyltestosterone **6** or methandienone **7** (Figure 2B).

T is not orally active as it readily undergoes hepatic metabolism (though there are some oral forms, such as undecanoate ester; attachment of a very long-chain ester at 17β position increases oral activity). The usual mode of administration includes injections or subcutaneous implants of its ester forms. Dermal patches are available and this is the method of choice for the treatment of hypogonadism [1,7]. T is also available in other therapeutic modalities, including topical hydroalcoholic gels [8], buccal [9], sublingual [10], and intranasal formulations [11].

Another compound that finds use in the management of low T levels is dihydrotestosterone (DHT, androstanolone; **4**, Figure 1). It is available as injections or dermal gels. Enormous efforts have been made to produce an orally active form of T, and one successful candidate is the undecanoate ester of T [12].

Methyltestosterone (**6**, Figure 2) is an orally active agent that is used for hypogonadism, erectile dysfunction, suppression of menopausal symptoms (hot flashes, osteoporosis, low libido), and in the treatment of breast cancer [13,14]. Mesterolone (8, Figure 3) has a 1α-methyl group and a reduced Δ^4^ double bond and is also orally active. Its androgenic activity is slightly higher than the anabolic effect, and it is of value for increasing low T levels, but it is hardly ever prescribed now [15,16]. Mesterolone has very low to no oestrogenic activity and shows only slight hepatotoxicity. The introduction of a methyl group in position 1α leads to an increased oral activity. Oral activity may also be achieved by the introduction of the 17α-alkyl group (as seen in methyltestosterone). This modification leads to reduced metabolism in the liver and increased bioavailability, but hepatotoxicity is also increased [17,18]. Methandriol (**9**, Figure 3) is available in both oral and injectable forms as dipropionate, propionate, and bisenanthoyl acetate esters. It has almost exclusively been used in the treatment of breast cancer in women [19,20].

All of the aforementioned derivatives have an androgenic to the anabolic activity ratio of about one to one. There have been attempts to produce steroids with low androgenic but high anabolic activity, but every anabolic steroid retains some androgenic activity. Anabolic activity may be increased by several chemical modifications, including introduction of a double bond between the C_1_ and C_2_ (e.g., metandienone, turinabol), between the C_9_ and C_10_ and the C_11_ and C_12_ positions (e.g., trenbolone, metribolone, tetrahydrogestrinone), introduction of a substituent such as a hydroxyl group or chlorine atom at the C_4_ position (e.g., turinabol), substitutions at the C_2_ or C_2_α position such as methyl (e.g., drostanolone), hydroxymethylene (e.g., oxymetholone), or a fused ring (e.g., stanozolol), and removal of the C_19_ methyl group (e.g., nandrolone, trenbolone, norethandrolone, ethylestrenol). Some of the agents with increased anabolic effects are described below.

Metandienone (dianabol^®^; **7**, Figure 2) is hardly ever used now in clinical practice [21]. It is available in both oral and injectable forms. Metandienone is a strong agonist for oestrogen receptors and can cause gynecomastia and fluid retention [22]. Many users are thus forced to take selective oestrogen receptor modulators (SERMs or SORMs), such as tamoxifen, to combat these side-effects [4]. Other side-effects include mental disorders, increased aggressiveness, and hepatotoxicity. Fluoxymesterone (halotestin; **10**, Figure 3) is a 17α-methyl-9α-fluoro-11β-hydroxy derivative. It is used in the treatment of hypogonadism, delayed puberty [23], female breast cancer [24], and anemia [25]. It can cause oedema because of sodium and water retention, presumably through inhibition of corticosteroid 11β-hydroxysteroid dehydrogenase enzymes [26]. It is still widely abused to improve strength and performance. Drostanolone (**11**, Figure 3) is another agent that has been removed from medicinal use, although it was of value in certain types of breast cancer [27]. Metenolone (**12**, Figure 3) has been used in the form of acetate and enanthate esters, the former being orally active, while the latter is given by injection. Both esters have been mainly used in the treatment of anemia caused by bone marrow failure [28]. Metenolone has weak androgenic and oestrogenic activity and low hepatotoxicity and has been discontinued for medicinal use in many countries. Oxandrolone (**13**, Figure 3) has a replaced carbon atom at the C_2_ position with an oxygen atom, which leads to reduced hepatotoxicity. It has the advantage of being primarily metabolized by the kidneys and not by the liver. It is especially useful for treating cases of severe weight loss and diseases that cause muscle wasting such as AIDS wasting syndrome, corticosteroid-induced protein catabolism, and alcoholic hepatitis [29], but also finds use in anemia, hereditary angioedema [30], severe burns, osteoporosis [31], hypogonadism, and Turner’s syndrome [32]. One of the most common side effects is a decrease in high-density lipoprotein (HDL). Oxandrolone has a high anabolic to androgenic activity ratio, which makes it especially suitable for use in women. It has low androgenic activity and relatively low hepatotoxicity. Oxymetholone (**14**, Figure 3) has a strong anabolic effect and is especially of value in treating anemia. It is also used in osteoporosis [33], AIDS wasting syndrome [34], and other conditions, in which muscle growth and weight gain are needed. The most common side effect is hepatoxicity [35]. When oxymetholone is treated with hydrazine, it forms a pyrazole ring fused to a saturated A ring. An example of such a compound is stanozolol [1]. Stanozolol (**15**, Figure 3) has been used in the treatment of osteoporosis [36] and is currently being evaluated as a treatment for hereditary angioedema [37,38]. Unlike other anabolic steroids, it is not available in esterified form but as an aqueous solution or tablets. The use of this drug in humans was discontinued in many countries, but it is still widely used in veterinary medicine for the same conditions as in humans. Boldenone (**16**, Figure 3) is a natural derivative of T, also known as Δ^1^-testosterone. It is available as a undecylenate ester and is exclusively used in veterinary medicine [16,39]. Among other activities, it increases appetite and also stimulates the release of erythropoietin. Boldenone has relatively low hepatotoxicity, androgenic potency and does not interact with receptors for progesterone. Turinabol (chlorodehydromethyltestosterone; **17**, Figure 3) is a 4-chloro derivative of metandienone. It was developed for the treatment of wasting diseases, especially for patients losing bone strength and mass [40].

In the course of synthesizing T analogues, compounds with the methyl group removed from position C_19_ were obtained and these displayed considerable progestogen activities. Nandrolone (19-nortestosterone; **18**, Figure 4) is available as decanoate and phenylpropionate esters, but these are not orally active and must be administered via subcutaneous or intramuscular injection. They are used in the treatment of anemia, severe burns, wasting syndrome in patients suffering from AIDS [41], osteoporosis [42], or breast cancer [39,43]. It is also available in the form of eye drops as nandrolone sulfate [44]. Ethylestrenol (**19**, Figure 4) lacks the 3-keto group. It was used for muscle promotion and weight gain, in treatment of bone pain and osteoporosis, as an adjunct therapy for corticosteroid-induced wasting and severe injuries, arthritis, aplastic anemia and anemia of chronic kidney disease [45,46,47], conditions of arteries and veins (e.g., thrombosis, Behçet’s disease, Raynaud’s disease, Degos disease) [48,49,50], and short stature in youths [51]. It is no longer used medicinally but is still available for veterinary use. Norethandrolone (**20**, Figure 4) has similar properties and is also used to treat muscle wasting [52], severe burns [53], and aplastic anemia [54]. Its medicinal use has largely been discontinued, though it is still used in some countries [53]. Trenbolone (**21**, Figure 4) has been marketed as a variety of esters, many of which are no longer used in veterinary or medicinal practice. Trenbolone acetate is still used in animals to stimulate muscle growth and appetite [55]. Perhaps the most common trenbolone ester for human use was hexahydrobenzylcarbonate, but this is no longer prescribed. A harmless, but potentially worrying adverse effect of trenbolone is an orange coloration of body fluids, including urine, because of the presence of a strong chromophore group. Another very specific side effect is the so-called trenbolone cough (particularly prevalent in trenbolone acetate), a phenomenon whose mechanism has not yet been satisfactorily explained, and is believed to be related to the interaction with prostaglandin receptors. Among other relatively common side-effects of trenbolone use are erectile dysfunction, reduced sex drive, night sweats, insomnia, increased aggressivity, anxiety, and cardiovascular problems [56].

Although research on novel anabolic steroids is quite limited, a few new androgenic-anabolic steroids are currently being developed. Trestolone (**22**, Figure 5) and dimethandrolone (**23**, Figure 5) are experimental compounds undergoing clinical testing as male contraceptives and in TRT for low T levels [57,58]. As with illegal drugs, some of the T derivatives were developed clandestinely as so-called “designer drugs”. These compounds have been chemically modified so that they are orally active and retain pharmacological activity and could be detected by standard anti-doping analytical tests only if monitored by the list of prohibited substances. However, the present World Anti-Doping Agency (WADA) list indicates that other substances with a similar chemical structure or similar biological effect(s) are prohibited. The efficacy and safety of the majority of designer steroids have not been properly evaluated in animal and human clinical trials, thus, the use of these drugs may lead to unexpected side effects. Some of these designer steroids are discussed below.

1-Testosterone (**24**, Figure 6 is a synthetic derivative of T having a Δ^1^ double bond instead of the Δ^4^ bond as in the natural molecule. It has both androgenic and anabolic activity and appears to be metabolized by the kidney [59]. Methasterone (**25**, Figure 6) is a 17α-alkylated orally active analogue of drostanolone. Methasterone was illicitly used as the main ingredient of a dietary supplement named Superdrol. It exhibits strong hepatotoxicity [60]. Desoxymethyltestosterone (DMT; **26**, Figure 6) is an orally active 17α-methylated derivative of dihydrotestosterone. It is unusual in having a Δ^2^ double bond and lacking the typical 3-keto group (compare with ethylestrenol). Animal studies have shown that DMT’s anabolic effects are stronger than androgenic activity. The most common side-effects are hepatotoxicity and cardiac hypertrophy [61]. Tetrahydrogestrinone (THG; 27, Figure 6) is an orally active agent, also known as ‘The Clear’. THG is a distinctive synthetic analogue with a methylated C_18_ residue and a system of three double bonds similar to trenbolone. Prolonged use of this compound may lead to infertility. Unlike most other anabolic steroids, THG binds to glucocorticoid receptors, which may result in serious complications due to weight loss. Another side-effect not seen with most other steroids is its potential immunosuppressive activity [62]. Norboletone (**28**, Figure 6) is another C_18_-methylated analogue. It was studied as an agent for use in the treatment of weight loss and short stature but concerns about its toxicity prevented it from being marketed as a pharmaceutical compound [63]. Metribolone (**29**, Figure 6) is a 17α-methylated derivative of trenbolone. It was quite extensively used in research as a ligand of the androgen receptor and a photoaffinity label. Metribolone was being also considered as an agent for advanced breast cancer in women but it has never been marketed for medicinal use because it is strongly hepatotoxic even at very low doses [64]. Methylstenbolone (**30**, Figure 6) is a more recent orally active agent that has never been approved for medicinal use but it has been used as an illicit dietary supplement [65,66].

### 2.2. Selective Androgen Receptor Modulators (SARMs)

SARMs are a novel group of compounds developed to selectively augment anabolic effects in muscles and bones, while avoiding undesirable androgenic effects in skin, larynx, and reproductive organs. The majority of these compounds lack the structural functionalities of the original anabolic steroids and are sometimes termed nonsteroidal androgens. It was hoped that these agents could be used in cases where conventional anabolic steroids produced undesirable side-effects, such as virilization in women and prostate hyperplasia in men [67]. Despite the enormous effort that has been expended in the development of selective anabolic agents, the androgenic effect is very hard to remove completely and many of the currently developed SARMs still do have some androgenic activity. None of the SARMs has been approved for therapeutic use as they are still in the clinical testing phase of development. Some SARMs are available on the black market and are being used recreationally. Since the safety of these agents is unknown, the use of SARMs may result in unexpected and very serious adverse effects such as the increased risk of myocardial infarction, stroke, and liver damage [68].

Enobosarm (**31**, Figure 7) is in clinical development for cancer cachexia, sarcopenia, breast cancer, osteoporosis, and stress urinary incontinence in menopausal women. However, it has failed in some clinical trials, e.g., for the treatment of wasting in patients with lung cancer nor was it shown to significantly reduce the stress incontinence episodes. Various adverse effects have been reported, including decreased blood glucose, headache and back pain, and reduced HDL [69,70,71,72,73]. Another SARM of interest is LGD-4033 or ligandrol (**32**, Figure 7). Its pharmacological properties are similar to enobosarm and it is also in the development phase for use in wasting syndrome and osteoporosis. Ligandrol was claimed to have hepatotoxicity and to alter HDL to low-density lipoprotein (LDL) ratio, and thus has the potential for increased risk of heart attack and stroke. More studies are required as there is still no definitive evidence [74,75]. Some companies illicitly marketed it as a dietary supplement. BMS 564929 (**33**, Figure 7) is currently in clinical development for the treatment of age-related loss of muscle mass, osteoporosis, metabolic syndrome, hypertension, reduced sex drive, and depression in men caused by a decline in T levels. Some adverse effects were observed in animal models, such as a reduction in luteinizing hormone level, but few side-effects are known for humans [76,77].

Various SARMs are currently in pre-clinical development, including AC-262356 [78], LGD-2226 [79], LGD-3303 [80], S-40503 [81], S-23 [82], S-1 [83], C-6 [84] and RAD140 [85] (**34**–**41**, Figure 7) [86]. Most of these agents are being tested for osteoporosis and wasting syndrome. S-23 and C-6 are in development as a male contraceptive drug. RAD140 is being investigated for the treatment of wasting syndrome and breast cancer.

Some SARMs have been removed from medicinal consideration because of not being orally active, problems with pharmacokinetics and pharmacodynamics, and/or due to unwanted adverse effects. These include acetothiolutamide [87], andarine (S-4) [88], LG-121071 [89], TFM-4AS-1 [90], MK-0773 [91], and YK-11 (**42**–**47**, Figure 8) [92,93].

### 2.3. Testosterone and Nandrolone Prodrugs (Prohormones)

T and nandrolone precursors, including androstenediol, androstenedione (**2** and **3**, Figure 1), 19-nor-5-androstenediol, 19-nor-4-androstenediol, 19-nor-5-androstenedione, and 19-nor-4-androstenedione (**48**–**51**, Figure 9), and more recently, DHEA (**1**, Figure 1) have been aggressively marketed for their ability to enhance T levels and build muscle mass [94]. Some of these compounds, especially DHEA, have also gained popularity for their alleged anti-aging properties, promoting wellbeing and youthful energy [1,95]. DHEA, androstenedione, and androstenediol are endogenous prohormones that are involved in the T biosynthesis. The synthetic variants, 19-norandrostenedione and 19-norandrostenediol, lack the 19-methyl group (compare with nandrolone) and are not found in nature [96]. In contrast to the marketing claims, studies have failed to confirm that long-term administration of T prohormones has any effect on serum T concentrations in healthy people. Among subjects with low endogenous T production, however, T prohormones were found to increase endogenous T levels, but only marginally. Interestingly, it was observed though that even a single dose of androstenedione (100 mg) has raised serum T level in women to a male reference interval [94]. Prolonged administration of androstenedione and androstenediol might raise oestrogen levels. There was also a suggestion that the 19-nor analogues of androstenediol and androstenedione are converted in the body to nandrolone, although there is little evidence to support this claim [97]. Chronic exposure to T prodrugs may lead to typical anabolic steroid side-effects including the development of cancer of the prostate, testes, or pancreas, gynecomastia, masculinization, reduced HDL, cardiovascular disorders, and aggressiveness [94]. As with anabolic steroids and SARMs. T prohormones are on the list of controlled substances in many countries, and their manufacture, possession, and use are prohibited.

### 2.4. Testosterone Boosters

T boosters are not strictly defined. They are preparations containing plant-derived ingredients that are supposed to increase T production or act as antioestrogens by inhibiting aromatase or oestrogen receptors. The elevation of T levels by the majority of these so-called boosters seems to be negligible. They are usually available in the form of legal dietary supplements, which is consistent with their lack of efficacy. Perhaps the most widely exploited T booster is derived from *Tribulus terrestris* (Zygophyllaceae), an annual plant indigenous to the Mediterranean region, warm temperate and tropical areas. The fruit powder and fruit extracts are used in the manufacture of *Tribulus* preparations. The T-boosting ingredient appears to be the furostane saponin, protodioscin (**52**, Figure 10). Although products containing *Tribulus* extract are heavily marketed for increasing T levels, there is little evidence that protodioscin or related compounds are converted in the body to T and they do not appear to elevate T levels. The *Tribulus*-containing preparations are not only used to increase strength and muscle mass but also in the hope of restoring pre-supplemental T levels after anabolic steroid withdrawal, though the results are far from those expected. Protodioscin is not marketed as a single agent. The plant material also contains alkaloids of the β-carboline type, namely harman, and norharman that can cause weakness or partial loss of voluntary movement of the extremities. Taking high doses of *Tribulus*-containing products may lead to the development of this side-effect. *Tribulus* is sometimes combined with underground parts of maca (*Lepidium meyenii*; Brassicaceae), a traditional Andean medicinal plant used in the treatment of sexual dysfunctions and to boost overall vitality. Much of the effect is believed to be associated with specific alkamides called macamides, which may increase sperm count and motility, but do not affect T levels [98].

Another widely marketed T-boosting product contains the powdered root or root extract of Mexican yams such as *Dioscorea villosa* (Dioscoreaceae). These plants produce tubers, which accumulate large amounts of bitter spirostane saponins, principally diosgenin, with smaller amounts of its 25β-epimer yamogenin (**53**, Figure 10). Though pharmaceutical T and other steroidal hormones are obtained through the chemical conversion of diosgenin (as indicated in Figure 2), there is no evidence that regular intake of diosgenin increases T levels in the human body. Supplements containing *Dioscorea* extracts are also taken to treat symptoms of menopause in women, as an alternative to hormone replacement therapy, but again, there is no definitive evidence that these saponins are metabolized to progesterone. Some T-boosters also contain fenugreek seeds (*Trigonella foenum-graecum*; Fabaceae), a spice that also contains saponins, diosgenin, and yamogenin [1,99,100].

The dried leaves of sisal (*Agave sisalana*; Asparagaceae) and the roots of sarsaparilla (various *Smilax* species; Smilacaceae) are occasionally used as purported T boosters. The major saponin of sisal is hecogenin (**54**, Figure 10) together with small amounts of tigogenin (**55**, Figure 10) and neotigogenin (**56**, Figure 10). Sapogenins present in sarsaparilla include smilagenin (**57**, Figure 10) and sarsasapogenin (**58**, Figure 10). These compounds all have the potential to serve as raw material for the synthesis of medicinally useful steroids. Users of these substances believe that they are metabolized to T in the human body, but clinical evidence does not support this experience [1,101].

An extract of maral root (*Rhaponticum carthamoides*; Asteraceae) and its major active component 20-hydroxyecdysone (**59**, Figure 11) has a long history of being used, especially by bodybuilders as a dietary supplement to increase protein synthesis and stamina. Animal studies as well as recent human trials support these claims. Participants administered with 20-hydroxyecdysone had significantly increased performance and muscle hypertrophy compared to controls. The mechanism of these effects seems to be associated with the ability of 20-hydroxyecdysone to interfere with β-oestrogen receptors. More studies are needed to clarify the exact mode of action. Maral root extract and 20-hydroxyecdysone are currently legal; however, 20-hydroxyecdysone was recently included in the WADA monitoring list [102].

Ashwagandha or Indian ginseng (*Withania somnifera;* Solanaceae) has recently received attention as a supplement that can increase muscle mass, strength, and overall fitness. The ground root contains several ergostane steroids termed withanolides, the major one being withaferin A (**60**, Figure 11). This compound has demonstrated various biological activities, among which an anticancer effect has shown some benefit in colon cancer cell models. The anticancer activity of withaferin A appears to be mediated in part by down-regulation of α-oestrogen receptor expression [103]. However, a link between withaferin A intake and increased T levels has not been observed. The ashwagandha products are sometimes combined with an extract of *Eleutherococcus senticosus* (Araliaceae), also known as Russian or Siberian ginseng, used as a substitute for the traditional ginseng, *Panax ginseng* (Araliaceae).

This plant is believed to have adaptogenic properties similar to those of ginseng and has been used in folk medicine to alleviate stress, but the mechanism is unknown. The root contains eleutheroside E, a phenylpropane glycoside, and several saponin-like structures based on sitosterol (eleutheroside A) and oleanolic acid (ciwujianoside E and eleutheroside M; **61**–**63**, Figure 11). *E. senticosus* appears not to have any significant effects on endogenous T levels [1,104,105].

Many lesser-known, steroid-producing plants are occasionally marketed as T-boosters, such as damiana (*Turnera diffusa*; Passifloraceae) or Malaysian ginseng (*Eurycoma longifolia*; Simaroubaceae). Damiana was reported to contain sitosterol 3-*O*-β-d-glucoside, while Malaysian ginseng has a lanostane derivative, tirucallane (**64** and **65**, Figure 11). Again, there is very little evidence that the compounds present in extracts of these plants increase endogenous T levels [106,107,108,109].

### 2.5. Antiandrogens

Antiandrogens comprise a relatively large group of compounds that interfere with the normal biological activity of T and various mechanisms have been proposed for their function. Some antiandrogens compete with androgens such as T for binding to androgen receptors, preventing receptor activation and the consequent biological effects. Androgen receptor antagonists include steroids (cyproterone, megestrol, chlormadinone, spironolactone, and oxendolone), nonsteroidal compounds (flutamide, bicalutamide, nilutamide, topilutamide, enzalutamide, and apalutamide), and progestins such as dienogest, drospirenone, medrogestone, nomegestrol, promegestone, and trimegestone. Other antiandrogens act by inhibiting the enzymes responsible for androgen synthesis. These include finasteride, dutasteride, epristeride, and alfatradiol, which block 5α-reductase activity and the so-called CYP17 inhibitors, ketoconazole, abiraterone, and seviteronel, which inhibit 17α-hydroxylase-17,20-lyase. Another group of antiandrogens, the antigonadotropins, suppress the production of gonadotropin-releasing hormone (GnRH) and reduce the secretion of follicle-stimulating hormone (FSH) and luteinizing hormone (LH; including leuprorelin, cetrorelix, various progestogens, and oestrogens). Compounds that stimulate sex hormone-binding globulin (SHBG) production raise levels of SHBG in the blood and decrease the availability of androgens such as ethinylestradiol and diethylstilbestrol). Anticorticotropins block the production of adrenal androgens by inhibition of adrenocorticotropic hormone (ACTH) and various glucocorticoids. Another antiandrogen strategy is the use of androstenedione immunogens to prepare a vaccine against androgen precursor androstenedione to generate antibodies that block androgen production (e.g., ovandrotone albumin and androstenedione albumin). The antiandrogen action of some agents is not restricted to one mechanism but may involve a combination of several. Some of these compounds have been used by athletes for masking steroid abuse. Additionally, some antiandrogens paradoxically display weak androgen and anabolic effects. For the sake of this review, only those antiandrogens that are based on the T structure are listed in this section. 

Cyproterone acetate (**66**, Figure 12) is a competitive androgen antagonist. It reduces male libido and fertility and is used to reduce high T levels, male hypersexuality, prostate cancer, early puberty, androgen-dependent skin and hair conditions (such as acne, hirsutism, female baldness), and in transgender hormone therapy. Cyproterone is relatively well tolerated, but patients may develop hypogonadism, infertility, osteoporosis, breast enlargement, gynecomastia, obesity, fatigue, depression, vitamin B_12_ deficiency, and reduced glucocorticoid activity. Cardiovascular problems, hepatotoxicity, and the development of some brain tumours were reported as rare adverse effects. Certain patients may develop withdrawal symptoms and adrenal insufficiency [110]. Finasteride and dutasteride (**67** and **68**, Figure 12) are both 4-aza-derived 5α-reductase inhibitors that find use in the treatment of prostate cancer, and enlarged prostate (benign prostatic hyperplasia). Continuous use of finasteride for up to six months is required to reduce an enlarged prostate, but the effects can last for up to twelve months after the drug is discontinued. Dutasteride seems to be a more effective agent for this disease, however. Both compounds are also indicated in male pattern baldness, excessive hair growth, and transgender hormone therapy. Common adverse effects include increased risk of high-grade prostate cancer due to the lowering effect on prostate-specific antigen (PSA), gastrointestinal distress, dizziness, headache, gynecomastia, and sexual dysfunctions [111]. Finasteride has also been used by athletes to mask steroid abuse. Abiraterone (**69**, Figure 12) is a CYP17 inhibitor and is especially useful in some types of prostate cancer, including castration-resistant and castration-sensitive variants (mCRPC and mCSPC, respectively). Its structure is related to that of pregnenolone, having a modified side-chain. Abiraterone use can cause tiredness, nausea, headache, hypertension, oedema, hypopotassemia, increased blood sugar, hot flashes, gastrointestinal discomfort, liver damage, and adrenocortical insufficiency [112]. Danazol and gestrinone (**70** and **71**, Figure 12) are classified as inhibitors of pituitary gonadotropin release but display other functions, including weak androgenic activity, inhibition of enzymes involved in androgen synthesis, and decrease of SHBG levels in the blood. They also act as antioestrogen and antiprogestogen agents and are particularly useful in the treatment of endometriosis to suppress the growth of endometrial tissue outside the uterus [1]. Various androgenic side effects were observed for both drugs, including acne, voice deepening, hirsutism, baldness, adverse blood lipid profiles, breast and clitoral enlargement, weight gain, fluid retention, and oestrogen deficiency. Gestrinone seems to provide less of these androgenic effects. Both are also of value in the reduction of uterine fibroids and menorrhagia. Danazol is also indicated in fibrocystic breast disease and hereditary angioedema [39,113].

## 3. Conclusions

T derivatives and many of its related analogues (AAS) have been primarily developed for medicinal use to treat various conditions (e.g., wasting syndromes associated with AIDS, anorexia, alcoholism, severe burns, muscle, tendon, and bone injuries, various types of anemias, and as a prophylaxis to hereditary angioedema). In due course, these drugs have been misused by athletes wishing to promote muscle development and strength and this phenomenon currently appears to be becoming considerably more frequent, especially in amateur athletes. Despite their abuse potential, these drugs are in some countries nonuniformly treated as controlled substances, while legal in others. Their legal status has also an impact on the availability of these drugs for medicinal use. Even though these agents are regarded as rather older generation drugs, and their efficacy/side-effects ratio may be in some cases viewed as disputable, they display valuable and often irreplaceable pharmacological properties, which makes them still medicinally useful. With the right type, dose, and appropriate regimen, AAS can be of value in the treatment of a relatively wide range of diseases and injuries where other drugs fail to provide the necessary therapeutic benefit. We should be especially careful with the more recent anabolic. Some SARMs are showing promising results in clinical trials, however, as of yet, they have not advanced into clinical use. Despite this, some SARMs are already appearing on the black market. T and the common steroidal anabolics have been used for more than fifty years, and we at least know what to expect from them. The new anabolics such as the SARMs could lead to unexpected and perhaps very dangerous side effects.

## Figures and Tables

**Figure 1 molecules-26-01032-f001:**
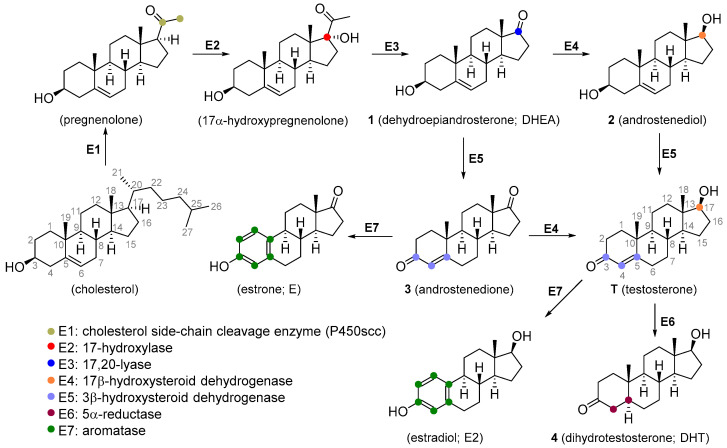
Scheme of testosterone biosynthesis.

**Figure 2 molecules-26-01032-f002:**
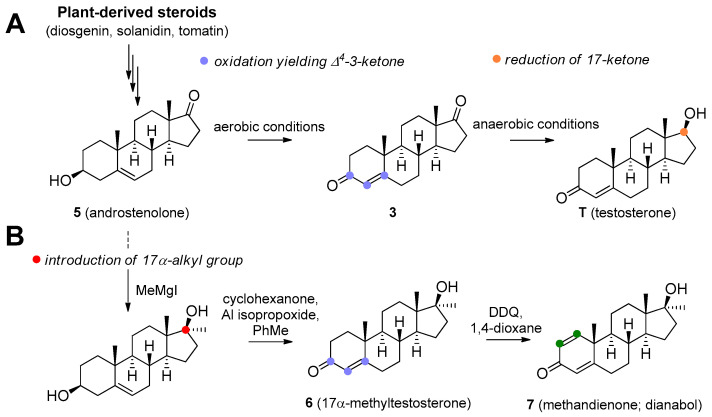
Microbial production of testosterone (**A**) and stereoselective introduction of alkyl (methyl) group to C_17_ position (**B**).

**Figure 3 molecules-26-01032-f003:**
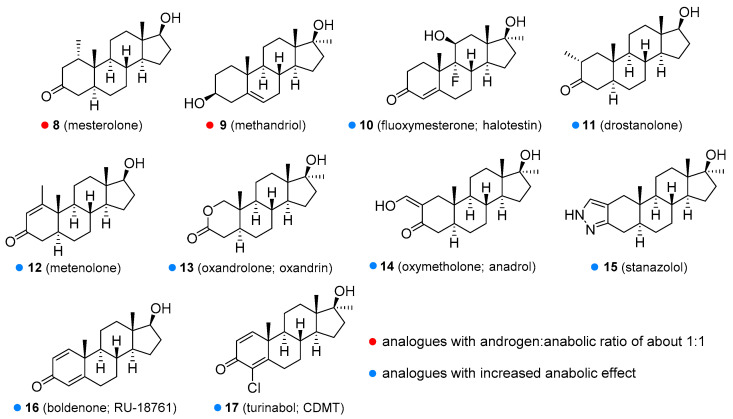
Some of the synthetic testosterone analogues (synthetic anabolic steroids).

**Figure 4 molecules-26-01032-f004:**
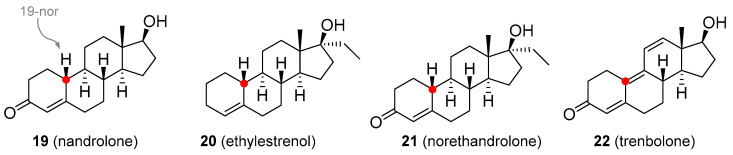
Nandrolone based analogues with a strong pro-estrogenic effect.

**Figure 5 molecules-26-01032-f005:**
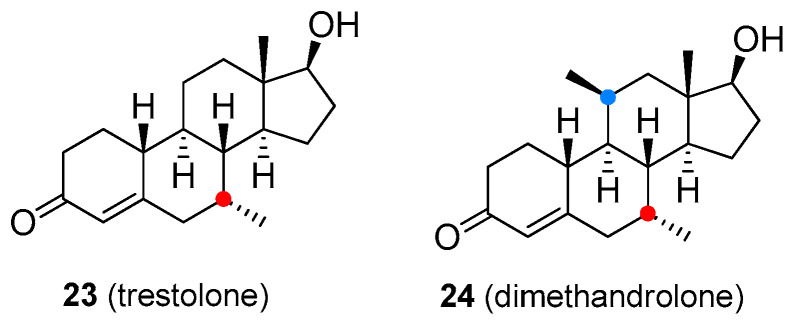
Experimental androgen-anabolic steroids.

**Figure 6 molecules-26-01032-f006:**
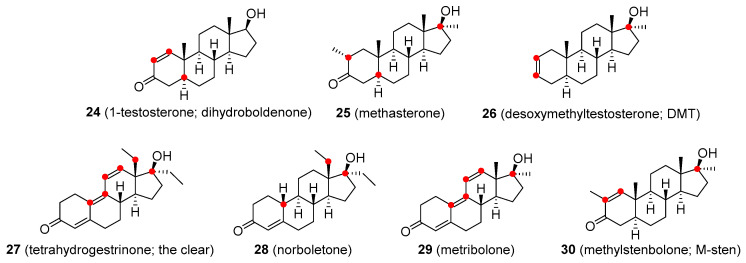
Designer drugs based on testosterone structure/functionalities.

**Figure 7 molecules-26-01032-f007:**
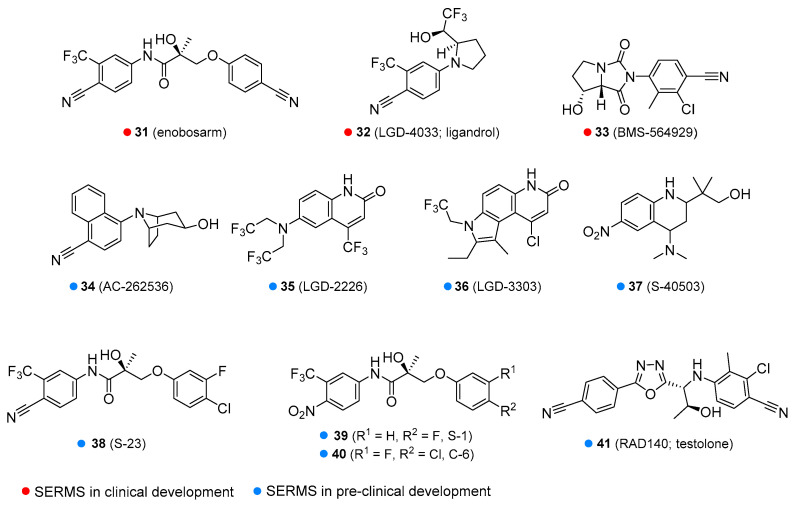
Some of the selective androgen receptors modulators in clinical or pre-clinical development.

**Figure 8 molecules-26-01032-f008:**
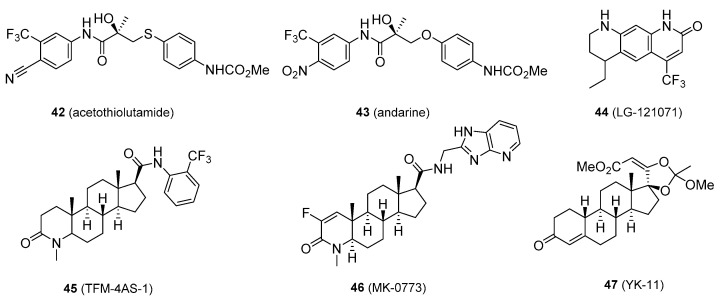
Selective androgen receptors modulators that have been eliminated from further research.

**Figure 9 molecules-26-01032-f009:**
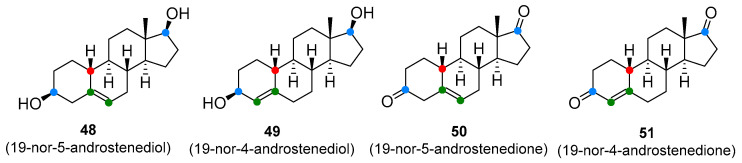
Testosterone prodrugs (prohormones).

**Figure 10 molecules-26-01032-f010:**
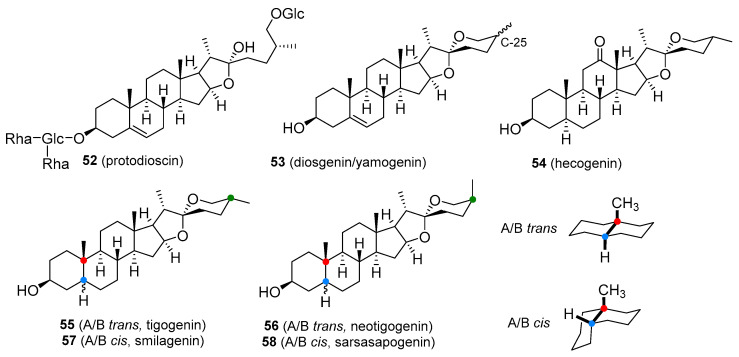
Plant-based steroids that are part of products marketed as testosterone boosters I.

**Figure 11 molecules-26-01032-f011:**
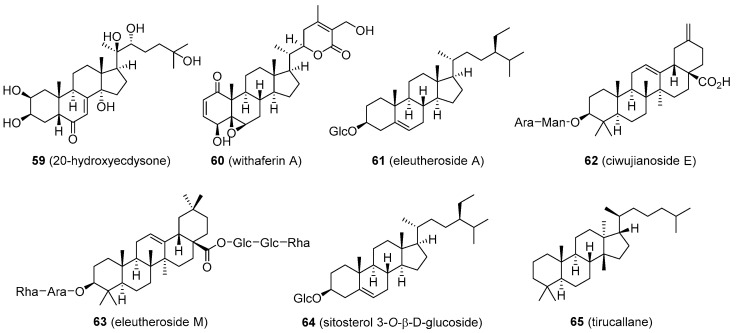
Plant-based steroids that are part of products marketed as testosterone boosters II.

**Figure 12 molecules-26-01032-f012:**
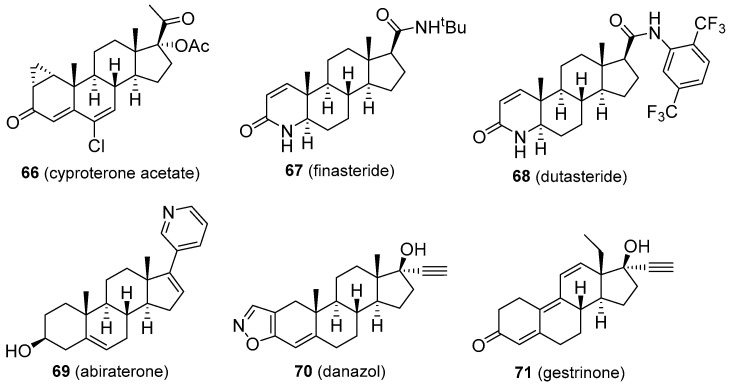
Molecular structures of anti-androgens.

**Table 1 molecules-26-01032-t001:** Medicinal use, usual routes of administration, and available forms of compounds discussed in this review.

Compound	Main Areas of Medicinal Application	Usual Route of Administration/Available Forms
**AAS**		
Testosterone	Male hypogonadism, oestrogen-dependent breast cancer in women, adjunct to hormone replacement therapy in menopausal women (to improve libido), testosterone replacement therapy (TRT)	Transdermal (patch, gel, cream), oral tablets (undecanoate ester), buccal, sublingual, and intranasal formulations, subcutaneous implants, and various esters for intramuscular injection (caproate, cypionate, decanoate, enanthate, isobutyrate, phenylpropionate, propionate, undecanoate)
Dihydrotestosterone (DHT; androstanolone)	Hypogonadism, gynecomastia, breast cancer (discontinued in some countries)	Transdermal gel, buccal and sublingual formulations, enanthate, propionate, and valerate esters (intramuscular injection)
Methyltestosterone	Delayed puberty, hypogonadism, cryptorchidism, erectile dysfunction, menopausal symptoms (osteoporosis, hot flashes, to improve libido), postpartum breast pain and engorgement, breast cancer in women	Oral tablets, buccal and sublingual formulations
Methandriol	Breast cancer in women (now discontinued in most countries)	Oral tablets or propionate and bisenanthoyl acetate esters (intramuscular injection)
Boldenone	Wasting syndrome, osteoporosis (now discontinued in most countries)	Undecylenate ester (intramuscular injection)
Fluoxymesterone (halotestin)	Hypogonadism, delayed puberty in males, breast cancer in women, some types of anemia	Oral tablets
Metandienone (dianabol^®^)	Hypogonadism (now discontinued in most countries)	
Drostanolone	Breast cancer (now discontinued in most countries)	Propionate ester (intramuscular injection)
Methenolone	Bone marrow failure-associated anemia, wasting syndromes, osteoporosis, sarcopenia	Acetate (orally active) and enanthate esters (intramuscular injection)
Oxandrolone	Osteoporosis-derived pain, weight loss, protein catabolism, AIDS-induced wasting, alcoholic hepatitis, severe burns, anemia, hereditary angioedema, Turner syndrome, hypogonadism, idiopathic short stature	Oral tablets
Oxymetholone	Anemia, osteoporosis (largely discontinued for these conditions), muscle wasting, AIDS wasting syndrome	
Stanozolol	Anemia, osteoporosis, burns, skeletal muscle injury (largely discontinued for these conditions), hereditary angioedema	Aqueous suspensions (intramuscular injection) or oral tablets
Turinabol	Osteoporosis (now discontinued in most countries)	Acetate ester (intramuscular injection) or oral tablets
Nandrolone	Burns, breast cancer, anemia, osteoporosis, HIV-induced wasting (now discontinued in most countries)	Decanoate and phenylpropionate esters (intramuscular or subcutaneous injections) or sulfate (eye drop formulation)
Ethylestrenol	Wasting syndromes, osteoporosis-associated pain, burns, severe injuries, various types of anemias, conditions of veins and arteries, arthritis, short stature (now discontinued in most countries)	Oral tablets
Norethandrolone	Wasting syndromes, burns, severe injuries, various types of anemias (now discontinued in most countries)	
Trenbolone	Wasting syndromes (now discontinued in most countries)	Acetate and hexahydrobenzylcarbonate esters (intramuscular injection)
**T and Nandrolone Prodrugs**		
Androstenediol	To raise T levels (never marketed for medicinal use)	Oral tablets
Androstenedione		
19-nor-5-androstenediol		
19-nor-4-androstenediol		
19-nor-5-androstenedione		
19-nor-4-androstenedione		
Dehydroepiandrosterone (DHEA)	To raise T levels (never marketed for medicinal use; though it is available as a dietary supplement)	
**Experimental AAS**		
Trestolone	Under development as a male contraceptive, for TRT, and hypogonadism	Subcutaneous implants, intramuscular injection (acetate ester)
Dimethandrolone	Under development as a male contraceptive and for TRT	Oral tablets
**Designer Steroids**		
1-testosterone	Never marketed for medicinal use	Intramuscular injections
Methasterone		Oral tablets
Desoxymethyltestosterone		
Tetrahydrogestrinone		Oral tablets, intramuscular injection
Norboletone		
Metribolone		
Methylstenbolone		
**SARMs**		
Enobosarm	Under development for cancer-related wasting, sarcopenia, breast cancer, osteoporosis, and stress urinary incontinence in menopausal women	Oral tablets
Ligandrol	Under development for wasting syndrome and osteoporosis	
BMS 564929	Under development for wasting syndrome, osteoporosis, diabetes, hypertension, reduced libido, depression	
AC-262356	In pre-clinical development	
LGD-2226		
LGD-3303		
S-40503		
S-23		
S-1		

NOTE: Since there is still little definitive (or no) evidence that testosterone boosters (products containing plant-derived extracts or isolated steroidal compounds) have an impact on endogenous testosterone levels and are not used medicinally (though available as dietary supplements), they are not included in this table.

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
