# Peer review of "Medicinal Use of Testosterone and Related Steroids Revisited"

_molecules, 2021, doi:10.3390/molecules26041032_

Round 1

Reviewer 1 Report

The manuscript entitled as " Medicinal use of testosterone and related steroids revisited " is suitable for publication. As the authors present the structures of the well known derivatives (androgens etc), I would suggest they make some comments about the SAR, eg methyltestosterone is a testosterone derivative that bears a methyl group at the 17 alpha position which  increases bioavailability. Minor spelling problems should be revised.

Author Response

Thank you very much for your review, please see the attachment.

Reviewer 2 Report

This manuscript describes the medicinal use of compounds based on the structure and biological activity of testosterone, and discusses some of the newer androgenic-anabolic compounds, such as selective androgen receptor modulators, whose efficacy/adverse-effect profiles have not been sufficiently established and may pose a greater risk than conventional androgenic-anabolic agents.

This subject is worthy of review and update. The manuscript is a good piece of work; it is well written and structured, and the information is clearly and succinctly presented. I suggest its publication in Molecules.

Authors should pay attention to the deformation errors on pages 104 and 360.

Author Response

(The authors gave the same response as above.)
